# Mitochondria in Human Fertility and Infertility

**DOI:** 10.3390/ijms24108950

**Published:** 2023-05-18

**Authors:** Jan Tesarik, Raquel Mendoza-Tesarik

**Affiliations:** MARGen Clinic, 18006 Granada, Spain; mendozatesarik@gmail.com

**Keywords:** mitochondria, sperm, oocyte, fertility, infertility, mitochondrial therapy

## Abstract

In human spermatozoa and oocytes (and their surrounding granulosa cells), mitochondria carry out important functions relating to human fertility and infertility. Sperm mitochondria are not transmitted to the future embryo, but are closely related to the generation of energy needed for sperm movement, capacitation, and acrosome reactions, as well as for sperm–oocyte fusion. On the other hand, oocyte mitochondria produce energy required for oocyte meiotic division and their abnormalities can thus cause oocyte and embryo aneuploidy. In addition, they play a role in oocyte calcium metabolism and in essential epigenetic events during the oocyte-to-embryo transition. They are transmitted to the future embryos and may thus cause hereditary diseases in the offspring. Due to the long life span of the female germ cells, the accumulation of mitochondrial DNA abnormalities often causes ovarian aging. Mitochondrial substitution therapy is the only way of dealing with these issues nowadays. New therapies based on mitochondrial DNA editing are under investigation.

## 1. Introduction

Mitochondria are known to be central players in energy production in adult cell types, providing the energy required for all cellular functions, although their multiple roles in cell metabolism of different specific cell types have been underestimated for a long time [1]. In addition to the well-known role of mitochondria as the powerhouse of the cell, the metabolic functions of mitochondria reach far beyond bioenergetics and include nutrient catabolism, the generation of biosynthetic precursors for macromolecules, the compartmentalization of metabolites for the maintenance of redox homeostasis, and metabolic waste management by acting as hubs [1].

Even less is known about the role of mitochondria in gametes and early embryos. Spermatozoa and oocytes are quite different cell types as compared with somatic cells. As for mature spermatozoa, the main function of mitochondria is to produce energy for their movement and fertilizing ability [2]. In addition to sperm movement, the sperm functions, depending on the correct function of mitochondria, include redox equilibrium, calcium regulation, and the control of apoptotic pathways, all of which are necessary for sperm capacitation, acrosome reactions, and sperm–oocyte fusion [2]. 

Many, though not all, of the problems caused by sperm mitochondrial abnormalities disturb spontaneous conception, but can be resolved with the use of assisted reproduction technologies (ARTs) and intracytoplasmic sperm injections (ICSIs), particularly because sperm-derived mitochondria are not transmitted to the embryos. 

On the other hand, oocyte mitochondria are transmitted to the early embryos, so that their abnormalities can have long-lasting consequences for further embryonic development and offspring health. Moreover, mitochondrial DNA (mtDNA) is more exposed to cumulated oxidative damage as compared to nuclear DNA because of its proximity to the sites of mitochondrial oxidative metabolism and the absence of DNA-associated proteins that can protect nuclear DNA against the oxidative damage. 

Here, we try to deal with this matter in view of the latest scientific achievements.

## 2. Roles of Mitochondria in Cell Metabolism

Even though the role of mitochondria in cell metabolism was long believed to be limited to the generation of energy needed for different cellular functions, more recent data point to other cell functions, such as the coordination of cellular adaptation to stressors, such as nutrient deprivation, oxidative stress, DNA damage, and endoplasmic reticulum stress [3]. Moreover, mitochondria are involved in intracellular calcium metabolism in some types of cells [4], coordinate cellular adaptation to stressors, oxidative stress, and DNA damage, in addition to producing metabolic precursors for macromolecules such as lipids, proteins, DNA, and RNA [1,3,5]. While the principal function of mitochondria in relation to energy production is the synthesis of adenosine trisphosphate (ATP), the sources from which ATP is produced vary, depending on the current aerobic/anaerobic condition and the type of each cell. Furthermore, in addition to ATP, mitochondria also produce metabolic precursors for lipids, proteins, DNA, and RNA, and also possess mechanisms to clear or utilize metabolic waste products [1]. 

## 3. Roles of Mitochondria in Germ Cells

Germ cells are highly specialized cells in which the roles of mitochondria are quite often sex-dependent. Therefore, mitochondria have somewhat different functions in spermatozoa and oocytes. 

### 3.1. Mitochondria in Spermatozoa

Sperm mitochondria, all of which are located in the midpiece of the sperm cell (Figure 1), are essential for sperm function by producing energy, maintaining redox equilibrium, and regulating calcium metabolism and apoptotic pathways, all of which are necessary for sperm movement, capacitation, acrosome reactions, and sperm–oocyte fusion (Figure 1). Each spermatozoon contains approximately 50–75 mitochondria, all of which accumulate in this sperm cell region. Even though sperm mitochondria are not transmitted to the future embryo [2], the abnormal function of sperm mitochondria can cause problems of spontaneous conception. Nonetheless, the use of assisted reproductive technologies (ARTs) can overcome these problems. In order to choose the best adapted ART method for each individual case, taking into account both the efficiency and cost-effectiveness, sperm characteristics, the wife’s age and ovarian reserve, and previous reproductive history are to be analyzed in more detail to obtain a synthetic view for each individual case and to adequately counsel the couples.

### 3.2. Mitochondria in Oocytes

Unlike sperm mitochondria, oocyte mitochondria do not only condition the oocyte quality, mainly in relation to the correct chromosome segregation during oocyte maturation (Figure 2), but they can also transmit maternally inherited abnormalities to the future embryo. In fact, mitochondrial dysfunction can not only jeopardize embryo viability, especially in older women, but also transmit maternally derived mitochondrial diseases to the progeny, independently of the mother’s age. In addition to generating energy required for basic cellular functions, through ATP synthesis, mitochondria also have the capacity to handle intracellular calcium homeostasis (a state of balance among different cellular compartments needed for the cells to survive and function correctly) through coordinated calcium storage and release; thus, they help to maintain free intracellular calcium levels required at each particular state of cell life. 

In addition to the above functions, mitochondria also take part in the control of oocyte and embryo epigenetic modifications (Figure 2) by regulating the most common molecular processes involved: histone acetylation, and histone and DNA methylation–demethylation [6,7].

Mitochondria are derived from the symbiosis of prokaryotic proteobacteria with the ancient archaea species and maintain some phenotypic features, such as the ability to produce ATP via a proton gradient created across their inner membrane [5]. Apparently, this kind of organelle is not developmentally adapted to endure a long life span. As compared to the short life span of spermatozoa (3 months of the whole spermatogenic process), oocytes are formed before birth and remain viable for several decades before fertilization. Correct mitochondrial function during this period depends on the proper interaction between the transcripts of the mitochondrial own DNA (mtDNA) and those of the nuclear genome of the oocyte, reflected by the normal oocyte and embryo competence [6]. 

mtDNA is more vulnerable to oxidative damage as compared to nuclear DNA because of the proximity of the sites of mitochondrial oxidative metabolism and the absence of DNA-associated proteins that protect nuclear DNA against the oxidative damage, whereby oocyte mtDNA damage appears to be the main factor of the decline in oocyte quality with female age, both in the physiological and premature ovarian aging [7]. Age-related mtDNA instability, which leads to the accumulation of mtDNA mutations in the oocyte, plays a key role in the deterioration of oocyte quality, in terms of the developmental competence of the future embryo and the risk of transmitting mitochondrial abnormalities to the offspring [7]. 

Interestingly, in addition to accumulating mtDNA damage, aging ovaries also suffer from the mitochondrial dysfunction of cumulus oophorus cells that surround oocytes [8]. In fact, at the antral follicle stage, ovarian granulosa cells differentiate into two populations: mural granulosa cells lining the wall of the follicle and cumulus cells surrounding the oocyte [8]. Since the cumulus cells regulate many functions in maturing oocytes, these dysfunctions appear to be closely related to the age-related decrease in human oocyte quality too [9].

In general, mitochondrial functions related to the success or failure of oocyte maturation and developmental competence, the appearance of chromosomal defects, fertilization failure, and early embryo demise in humans include the ability of mitochondria to balance ATP supply for critical developmental events during this period (Figure 2), and these functions are closely related to the mtDNA copy number, the timing of mtDNA replication during oocyte maturation, and functional compartmentalization whereby ATP may be preferentially supplied to those oocyte regions in which critical developmental processes occur at the required time [10].

There are many aspects of human oocyte function that remained unexplained until quite recently. First of all, human primordial follicles, with their respective oocytes, are formed during fetal development and remain dormant in the ovary for up to 50 years. There are still questions about how the oocyte mitochondria retain the ability to support the development of a new organism, even though the oocytes remain metabolically active during dormancy, and this must maintain mitochondrial activity for the biosynthesis of essential biomolecules [1]. The formation of reactive oxygen species (ROS), normally required for energy formation in mitochondria, is also a problem for mitochondrial longevity, given that their formation occurs in a close vicinity of mtDNA and given that mtDNA is not protected against oxidative damage by associated proteins, unlike the case of nuclear DNA, leading to age-related oocyte damage [11]. This question is partly explained by a recent study which shows that the oocyte mitochondria of long-living species escape the oxidative mitochondrial damage caused by ROS, since the oocyte can evade excessive ROS formation through remodeling the mitochondrial electron transport chain via the elimination of complex I [12]. The mitochondrial complex I, also known as respiratory complex I or NADH:ubiquinone oxidoreductase, is the first large protein complex involved in the respiration of many organisms from bacteria to humans. Its suppression during the dormant state of human oocytes, when the requirements for cell respiration are limited, can help to resolve this problem.

Even under these special conditions, mtDNA progressively accumulates mutations and deletions over time in different mammalian species, including humans, and these mtDNA defects negatively impact oocyte function and reproductive ability [13]. In addition to mtDNA lesions, oocyte aging is also accompanied by a reduced mtDNA copy number, leading to insufficient ATP levels and subsequent infertility and abnormal embryo development, and, in some cases, mitochondrial diseases in the offspring, often associated with oocyte aneuploidy [13]. In addition, oocyte viability and developmental potential are highly dependent on the function of both mural granulosa cells and cumulus oophorus cells [8,9], and there is growing evidence which suggests that abnormal mitochondrial function in these cell types can be a decisive element in causing age-related oocyte decay [14,15]. 

## 4. The Role of Mitochondria in Embryos

As mentioned above, all mitochondria that are active in human embryos are of oocyte origin. Thus, maternal mitochondrial abnormalities are inevitably transmitted to the embryos resulting from their fertilization. mtDNA codes for 13 essential subunits of the respiratory chain complexes, whereas all the other mitochondrial proteins, approximately ~1500, are nuclear-encoded [7]. If mitochondrial demise is merely a reflection of ovarian aging, it can cause repeated implantation failures and spontaneous abortions. On the other hand, if mitochondria carry specific maternally inherited mutations, compatible with implantation, pregnancy, and birth, these mutations will be transmitted to the offspring, leading to the transmission of corresponding diseases that often cause the death of the offspring in early childhood, and this can occur independently of maternal age. Leigh syndrome, characterized by 8993T>G mutation in subunit 6 of the ATPase gene [15], which could be successfully resolved by mitochondrial substitution therapy [16], is an example of the latter condition. In that pioneering study, the authors reported a case of a 36-year-old asymptomatic woman, a female carrier of Leigh syndrome with a history of four pregnancy losses (between 6 and 16 weeks of gestation), multiple undiagnosed pregnancy losses, and two deceased children (at ages 8 months and 6 years), presumably as a result of this disease. After the mother’s chromosomes were transformed into enucleated donor oocytes, an IVF (ICSI) of the resulting reconstructed oocytes resulted in a pregnancy and the delivery of a healthy boy [16]. 

In addition to Leigh disease, there are a number of other rare primary mitochondrial diseases caused by defects in mtDNA-encoded genes [16], most of which cause ophthalmologic manifestations. Even though most of these diseases are not lethal and are individually uncommon, they collectively pose a significant burden on human health and jeopardize the wellbeing of affected individuals. It remains to be determined if mitochondrial replacement therapy, such as that successfully used for Leigh syndrome [16], would be effective in other mitochondrial diseases too. 

It is well known that oocyte mitochondria dysfunction plays an essential role in oocyte aging [11,17]. In addition to oocyte mitochondrial aging, age-related deterioration also occurs in both mural granulosa cells and oocyte-associated cumulus granulosa cells and plays an important, though indirect, role in the decrease in oocyte and embryo quality [11]. In fact, granulosa cells need energy to transform androgens, produced by the theca interna cells into estrogens, according to the generally accepted two-cell two-gonadotropin theory which postulates that luteinizing hormones (LHs) stimulate the cells of the theca interna of ovarian follicles to produce androgens, whereas follicle-stimulating hormones (FSHs) stimulate the transformation of androgens into estrogens [18]. The prevalence of androgens, resulting from their inefficient transformation into estrogens, disturbs the progression of oocyte cytoplasmic maturation. In fact, estrogens and androgens exert antagonistic effects on human oocyte cytoplasmic maturation, acting via non-genomic mechanisms through specific receptors located on the oocyte surface [19,20]. 

There is also accumulating evidence which suggests that mitochondria contribute to epigenetic regulation in oocytes and embryos [17]. It is well known that epigenetic regulations, mediated by dynamic changes in oocyte and early embryo nuclear DNA methylation status (Figure 2), are crucial for correct genomic imprinting and normal embryo development [21,22]. Mitochondrial metabolites are important intermediates whose deficiency can disturb mitochondrial nuclear communication required for the generation and modification of nuclear epigenetic marks [17]. 

## 5. Mitochondrial Therapies for Human Infertility

### 5.1. Mitochondrial Replacement Therapies

Since most sperm mitochondrial abnormalities can now be resolved using ICSIs for fertilization, the treatment of oocyte mitochondrial abnormalities is the principal challenge for current research. At the time being, only mitochondrial substitution therapies are clinically available. They are based on the enrichment of the mitochondrial population of the patients’ oocytes with mitochondria from healthy young oocyte donors. Basically, two different techniques were developed in the late 1990s: one using oocyte cytoplasmic transfer from donor oocytes to patients’ oocytes [23,24,25] and the other based on the transfer of oocyte chromosomes attached to the meiotic spindle from the patients’ oocytes to previously enucleated donor oocytes [26]. Interestingly, both techniques yielded similar clinical outcomes in humans, and healthy mitochondria were detected in the offspring [27], although the proportion of healthy mitochondria is clearly lower after oocyte cytoplasmic transfer as compared with the chromosome transfer [26], probably because of a purifying selection of healthy mitochondria in the injected oocytes, leading to the faster proliferation of healthy mitochondria as compared to the abnormal ones [28]. 

The chromosome–spindle transfer technique [26] has been successfully used abroad by clinics from the US and Spain, two countries in which the technique had still not received official permission from the local administration. In the former case, it was used to avoid the maternal transmission of a known mtDNA mutation 8993T > G, known as Leigh syndrome. The treatment, consisting of the transfer of the patient’s metaphase chromosomes into previously enucleated donor oocytes, was carried out in Mexico and resulted in the birth of a healthy boy [19,29]. 

In brief, the female patient was a 36-year-old asymptomatic woman with a history of four pregnancy losses (between 6 and 16 weeks of gestation, reasons unknown) and two deceased children (at ages 8 months and 6 years) caused by Leigh syndrome, as confirmed by a mutation load of over 95% [19].

In the other case, the treatment was performed in couples with idiopathic infertility, without any previous detection of mitochondrial disease, by a mixed Spanish–Greek team in Greece, resulting in the birth of six healthy children [30]. The study involved twenty-five infertile couples with multiple previous unsuccessful IVF attempts (3–11), no previous pregnancy, no confirmed history of mitochondrial disease, and a female age of <40 years. Their previous IVF attempts were characterized by a pattern of low fertilization rates and/or impaired embryo development. Couples with severe male factor infertility were excluded [30].

It has to be admitted that, in cases with a history of idiopathic infertility, in which no clearly defined mitochondrial abnormality was detected, the success of both cytoplasmic transfer from young donors and chromosome–spindle transfer from the patients’ oocytes into previously enucleated donor oocytes is not necessarily due merely to mitochondrial replacement, since the oocyte cytoplasm, in addition to mitochondria, also contains many other developmentally important factors, mainly stored maternal mRNA species that are used for embryonic development before the activation (relatively late in humans) of the embryonic gene’s own transcription and expression (see above). Notwithstanding, from a clinical point of view, current information on the success of these treatments, without any signs of adverse effects for embryo and offspring health, should encourage the use of these techniques in the clinical practice of infertility treatment and make national administrations of different countries rethink the reasons why those techniques are not possible in their own countries. The corresponding decisions should take into account the level of stress and additional cost to be covered by the patients forced to visit other countries for interventions that could be carried out in their own ones.

A recent clinical study confirmed that mitochondrial substitution therapy using mitochondria obtained from autologous oogonia stem cells rather than donor oocytes is another effective clinical option to enhance embryo quality in recurrent fertilization failure cases [31]. In fact, there had been previous reports on mitochondrial therapy using mitochondria from oogonia stem cells, but this was the first time that the status of children born with the help of this technology was referred to. The authors isolated putative oogonial stem cells (p-OSCs) from the adult patients’ ovarian cortical tissue and extracted autologous mitochondria from them. The total number of p-OSCs collected from each patient (n = 52) ranged from 11,327 to 664,000 (mean: 172,778) [31]. Mitochondria were extracted from frozen p-OSCs on the day of fertilization via ICSIs, according to the patented protocol.. Patients’ oocytes were retrieved from large ovarian follicles using standard protocols of ovarian stimulation and ovarian puncture [31]. After centrifugation, approximately 2 pL of mitochondrial suspension and a spermatozoon were injected together into each oocyte [31]. Overall, 702 mature oocytes from 52 women aged 27–49 years were treated, resulting in a 61.5% fertilization rate and a 23.8% clinical pregnancy rate per embryo transfer. The embryo transfers resulted in 11 live births (3 sets of twins and 8 singletons), 1 other intrauterine fetal death (1 fetus of a twin pregnancy), and 4 miscarriages. The average implantation rate and birth rate were 18.6% and 17.5%, respectively [31]. The physical and cognitive development of all the babies born was normal, and no evidence on mtDNA heteroplasmy was found [31].

### 5.2. Mitochondrial Genome Editing 

Though still not available for clinical use, new therapies using mitochondrial genome editing [32], which would avoid both the third-party (oocyte donor) participation and the discomfort related to the patient’s oogonia stem cell recovery, are currently under investigation. It is important to accommodate existing genome editing techniques that are successfully used for nuclear DNA and to make them effective in the case of mtDNA. 

With the development of nuclear gene editing technology, the question now arises regarding how to target their actions to mtDNA instead of nuclear DNA. First demonstrated in 2012 [32], CRISPR/Cas9 genome editing technology became a revolutionary tool for nuclear gene editing. Further research suggests that Cas9 can be easily programmed by a guide RNA (gRNA) to act in different cellular targets. Hence, extensive efforts have been devoted to optimizing this system in order to eliminate mutant mtDNA [32]. Though encouraging, these results still need to be validated by future studies before being considered applicable in human reproductive medicine [32].

### 5.3. Currently Available Conservative Therapies

Mitochondrial replacement therapies are safe and efficient. Despite that, they are still banned in many countries, without any scientifically based reason. For couples who decide to carry out these therapies in other countries, this decision requires an additional cost for travel and accomodation, in addition to emotional discomfort. In this situtation, the use of previously validated conservative mitochondrial therapies should be taken into account. One of the most efficient oral treatments is melatonin, a natural hormone which combines the activities of a direct antixoxidant agent and those of a modulator of systems that protect cells against oxidative stress, making it a first-line treatment for premature or physiological ovarian aging [10,14]. Other antioxidants, such as vitamins C and E, as well as coenzyme Q10, can also be of help [10,14]. All of these treatments have the advantage of being easily available and applicable (oral use), and should be attempted before more complicated and controversial treatment options are resorted to.

## 6. Conclusions

Both sperm and oocyte mitochondria have specific functions in human fertility. Sperm mitochondria are essential for sperm fertilizing ability by providing energy required for adequate sperm movement, capacitation, acrosome reaction, and sperm–oocyte fusion. However, they are not transmitted to the future embryo. Hence, if fertilization is assisted by micromanipulation techniques, their abnormalities do not need to disturb further embryo development. However, this does not mean that all infertile couples with abnormal sperm characteristics should be directed indiscriminately to different types of ART. In fact, substantial improvements in sperm function due to mitochondrial disorders can sometimes be achieved via the oral treatment of men with different combinations of antioxidants, of which melatonin appears to be the most effective one. If there is no emergency in relation to female age and ovarian function, the couples can try and conceive spontaneously after the male undergoes antioxidant treatment. In other cases, the treatment effects on sperm quality should be evaluated after 2–3 months, and the least invasive and unexpensive ART technique should be attempted first, before proceeding to more invasive and expensive ones.

On the other hand, oocyte mitochondria, in addition to generating energy for basic cellular processes, also influence oocyte/embryo epigenetic events and are transmitted to the embryo. Consequently, their abnormalities may have long-lasting negative effects on embryo development. As in the case of spermatozoa, oral anti-oxidants, mainly melatonin, can have some positive effects in some cases. However, in view of the fact that the lifespan of oocyte mitochondria is much longer as compared to that of sperm mitochondria, oocyte mtDNA defects, accumulated over time, are usually less responsive to this conservative type of treatment. Mitochondrial replacement therapies, using either mitochondria from donor oocytes or those extracted from autologous oogonia stem cells, are the most effective treatment options available nowadays. Studies that aim to develop novel therapies based on mtDNA gene editing are in progress.

## Figures and Tables

**Figure 1 ijms-24-08950-f001:**
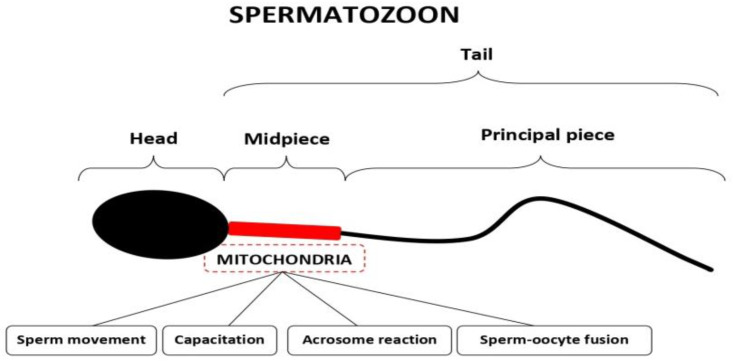
Schematic representation of a human spermatozoon showing the location of mitochondria and outlining their roles in sperm function and fertilization.

**Figure 2 ijms-24-08950-f002:**
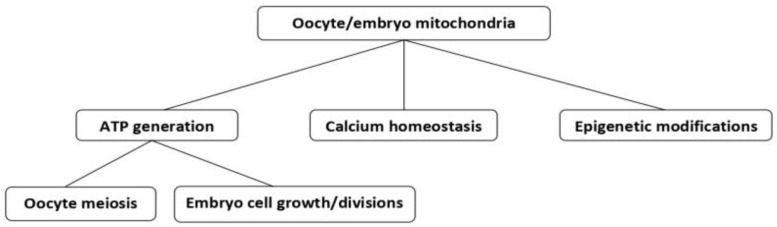
Different functions of mitochondria in human oocytes and embryos (see the main text for more explanations).

## Data Availability

Not applicable.

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
