# Peer review of "Mitochondria in Human Fertility and Infertility"

_ijms, 2023, doi:10.3390/ijms24108950_

Round 1
Reviewer 1 Report
The opinion is focused on a summary of findings on the function and dysfunction of mitochondria in gametes and the early embryo.
It is written clearly and legibly. However, I am missing some summary information and I have the following recommendations:
1) There is no mention of the involvement of mitochondria in steroidogenesis (lines 38-45).
2) Information "... also produce metabolic precursors for lipids, proteins, DNA, and RNA..." is repeated already on line 48.
3) I lack a description of the location of mitochondria in sperm cell, while this is interesting and how many there are; please discuse the number of mitochondria in the oocyte and during oocyte maturation.
4) I also lack an explanation for readers why sperm mitochondria transfer to the embryo does not occur or why sperm mitochondria are eliminated in the oocyte and how.
5) Please enlarge the figures, the font is not legible.
6) The sentence (lines 69-72) is taken out of context, it is general information about mitochondrie. I suggest to move it to the Introduction.
7) The links to Fig. 2 in the text do not completely correspond to what is in the picture. It would be better put Fig. 2 at the beginning of the chapter on oocytes and write a few sentences about it.
8) Some terms in the Fig. 2 are not explained and referred in the text, e.g. calcium homeostasis.
9) I'm missing examples of diseases linked to mitochondria in the chapter on the embryo, and in addition, please give an example of diseases causing offspring death (line 130).
10) Lines 134-135 again refer to the oocyte, similarly lines 138-147. I suggest to move these senteces to the oocyte chapter with link to the embryo.
Author Response
Responses to the Reviewer’s Comments
The opinion is focused on a summary of findings on the function and dysfunction of mitochondria in gametes and the early embryo.
It is written clearly and legibly. However, I am missing some summary information and I have the following recommendations.
Response: We have added a short summary after the Abstract and before the Keywords.
1) There is no mention of the involvement of mitochondria in steroidogenesis (lines 38-45).
Response: We have added a citation (Reference 11) of a study dealing with the role of mitochondria in steroidogenesis.
2) Information "... also produce metabolic precursors for lipids, proteins, DNA, and RNA..." is repeated already on line 48.
Response: All unnecessary duplications have been removed.
3) I lack a description of the location of mitochondria in sperm cell, while this is interesting and how many there are; please discuse the number of mitochondria in the oocyte and during oocyte maturation.
Response: We have added information about the location of mitochondria in sperm cells and their approximative number per sperm cell and oocyte, together with the corresponding citations (References 5 and 17).
4) I also lack an explanation for readers why sperm mitochondria transfer to the embryo does not occur or why sperm mitochondria are eliminated in the oocyte and how.
Response: This is an intriguing question. However, no studies have explained the corresponding mechanism yet.
5) Please enlarge the figures, the font is not legible.
Response: We have enlarged the figures as suggested.
6) The sentence (lines 69-72) is taken out of context, it is general information about mitochondrie. I suggest to move it to the Introduction.
Response: We have transferred this sentence to the Introduction (highlighted)
7) The links to Fig. 2 in the text do not completely correspond to what is in the picture. It would be better put Fig. 2 at the beginning of the chapter on oocytes and write a few sentences about it.
Response: We have put Fig. 2 at the beginning of this chapter and added some eplanations there (highlighted).
8) Some terms in the Fig. 2 are not explained and referred in the text, e.g. calcium homeostasis.
Response: In the first paragraph of the section 2.2., we have added an explanation of how mitochondria participate in the balanced metabolism of calcium (calcium homeostasis) and in the oocyte and embryo epigenetic modifications (highlighted).
9) I'm missing examples of diseases linked to mitochondria in the chapter on the embryo, and in addition, please give an example of diseases causing offspring death (line 130).
Response: Some examples of mitochondrial diseases that can cause severe offspring disease or (rarely) death have been added here (highlighted).
10) Lines 134-135 again refer to the oocyte, similarly lines 138-147. I suggest to move these senteces to the oocyte chapter with link to the embryo.
Response: The corresponding modification has been made.
Reviewer 2 Report
Comments about the manuscript:
“Mitochondria in human fertility and infertility”
The mitochondria of sperm and oocytes act on human fertility. Sperm mitochondria, which are not transmitted to the future embryo, are involved in the movement and capacitation of sperm, in the acrosome reaction and in the fusion of sperm with the oocyte. Oocyte mitochondria are involved in meiosis, they act on calcium metabolism and are involved in epigenetic phenomena. The abnormalities of mitochondria can be transmitted to the embryos and be responsible for hereditary diseases. The authors of this manuscript consider that mitochondrial replacement therapy represents the only way to treat these diseases. For this, studies are focusing on new therapies based on mitochondrial DNA.
This manuscript concerns the importance of sperm and ovum mitochondria at fertilization and during embryonic development. The authors summarize successively the roles of mitochondria in cell metabolism, the roles of mitochondria in germ cells, the role of mitochondria in embryos. They conclude by giving their opinion (the only way to treat diseases arising from germ cell mitochondrial DNA is mitochondrial replacement therapy).
This text is well written, clear and well documented. Whether or not we agree with the authors, their opinion is well supported. For me, the text could be published as is. I have only one minor comment.
Page 4, line 140. “two-cel-two-gonadotropin theory”: write “two-cell-two-gonadotropin theory”.
Author Response
Page 4, line 140. “two-cel-two-gonadotropin theory”: write “two-cell-two-gonadotropin theory”.
Response: This typing error has been corrected (highlighted).
The report states that the revised manuscript is acceptable in its current form.
Round 2
Reviewer 1 Report
Tha authors satisfactorily completed the manuscript on the basis of my comments. In my opinion, it is suitable for publishing now.
Author Response
The report states that the revised manuscript is acceptable in its current form.